# “Air Sign” in Misdiagnosed Mandibular Fractures Based on CT and CBCT Evaluation

**DOI:** 10.3390/diagnostics14040362

**Published:** 2024-02-07

**Authors:** Michał Gontarz, Jakub Bargiel, Krzysztof Gąsiorowski, Tomasz Marecik, Paweł Szczurowski, Jan Zapała, Grażyna Wyszyńska-Pawelec

**Affiliations:** Department of Cranio-Maxillofacial Surgery, Jagiellonian University Medical College, 30-688 Cracow, Poland; jakub.bargiel@uj.edu.pl (J.B.); krzysztof.gasiorowski@uj.edu.pl (K.G.); tomasz.marecik@uj.edu.pl (T.M.); pawel.szczurowski@uj.edu.pl (P.S.); jan.zapala@uj.edu.pl (J.Z.); grazyna.wyszynska-pawelec@uj.edu.pl (G.W.-P.)

**Keywords:** mandibular fractures, diagnostic errors, cone-beam computed tomography, computed tomography, missed diagnosis

## Abstract

Background: Diagnostic errors constitute one of the reasons for the improper and often delayed treatment of mandibular fractures. The aim of this study was to present a series of cases involving undiagnosed concomitant secondary fractures in the mandibular body during preoperative diagnostics. Additionally, this study aimed to describe the “air sign” as an indirect indicator of a mandibular body fracture. Methods: A retrospective analysis of CT/CBCT scans conducted before surgery was performed on patients misdiagnosed with a mandibular body fracture within a one-year period. Results: Among the 75 patients who underwent surgical treatment for mandibular fractures, mandibular body fractures were missed in 3 cases (4%) before surgery. The analysis of CT/CBCT before surgery revealed the presence of an air collection, termed the “air sign”, in the soft tissue adjacent to each misdiagnosed fracture of the mandibular body. Conclusions: The “air sign” in a CT/CBCT scan may serve as an additional indirect indication of a fracture in the mandibular body. Its presence should prompt the surgeon to conduct a more thorough clinical examination of the patient under general anesthesia after completing the ORIF procedure in order to rule-out additional fractures.

## 1. Introduction

Fractures of the mandible are prevalent injuries in the maxillofacial region addressed by maxillofacial surgeons. The main contributing factors include vehicle accidents, assaults, and falls, as well as incidents involving bicycles or electric scooters [1,2]. The primary cause of adult mandible fractures in the United States stems from interpersonal violence, predominantly among men aged from 18 to 24. Research analyzing 13,142 patients highlighted that men face a significantly higher risk, with almost half of these fractures resulting from assaults, often associated with alcohol consumption [3]. On the other hand, women more frequently experience mandibular fractures due to motor vehicle accidents (MVAs) and falls [3,4,5]. Approximately a quarter of women’s mandible fractures result from falls, but it is crucial to investigate potential cases of domestic violence if the injury does not align with accidental trauma. Recent advancements in safety technology for passenger vehicles have altered the injury trends and epidemiology of facial fractures in developed countries. Studies indicate that the use of seatbelts in conjunction with airbags can reduce the likelihood of sustaining a facial fracture during an MVA by more than 50% [6,7,8]. Mandibular fractures make up from 40% to 62% of facial bone fractures, with over 50% of these cases involving multiple fracture lines [9].

Mandibular fractures are diagnosed based on a combination of clinical examination and radiographic assessment. The critical factor in the diagnostic workup of mandible fractures is the evaluation of the patient’s occlusion. During the clinical examination, the physician ought to palpate the fracture site using bimanual techniques to assess fragment mobility. If there is no mobility, it suggests a stable fracture that might be treatable through conservative methods, provided the occlusion remains unchanged. Additionally, it is of great importance to conduct a careful observation and examination of intraoral lacerations, soft tissue injuries, and hematomas present around the fracture site. These factors carry the potential to significantly increase the susceptibility to infections, thereby necessitating thorough attention and monitoring. Typical signs such as the presence of ecchymosis on the floor of the mouth frequently serve as key markers for identifying mandibular fractures. An integral component of the evaluation involves a comprehensive assessment of the patient’s dental condition, which holds significant importance in the overall diagnostic process [10].

Radiographic assessment is very important in evaluating patients with significant facial trauma. Many patients with mandible fractures, especially in cases of multiple injuries after MVAs, arrive at the emergency room and typically undergo polytrauma computed tomography (CT) imaging to check for cervical spine (C-spine) and other associated injuries [10]. However, diagnostic errors constitute one of the reasons for the improper and often delayed treatment of mandibular fracture, leading to a higher risk of complications such as infection, malunion, nonunion, or osteomyelitis [11]. The inaccurate interpretation of radiographic images has consistently posed a significant challenge in preoperative diagnostics. For example, during the 1980s, in the realm of imaging diagnostics relying on conventional panoramic tomography and oblique projection of the mandible, the incidence of false-negative diagnoses was 19% among radiologists and 24% within the clinician group [12]. Until the early 2000s, panoramic imaging held the status of the preferred method for diagnosing mandibular fractures, achieving an 86% diagnosis rate and being a cost-effective procedure [13]. Nevertheless, CT scans now offer a thorough perspective of the mandible in the axial, coronal, and sagittal planes, along with the capability of producing three-dimensional reconstructions. With an almost 100% diagnostic rate, CT scans have become an indispensable tool in the diagnosis of mandibular fractures [13]. CT has the highest accuracy for the assessment of mandibular bone continuity and the detection of direct radiographic signs of fracture. However, it should be noted that mandibular body fractures are open fractures, which may imply the presence of air in the soft tissues at the fracture site, which can be seen on CT/CBCT scans. Until now, radiologic diagnosis has not taken into account the indirect sign of a mandibular body fracture in the form of air bubbles in the soft tissues.

The primary goal of treating mandibular fractures is to restore the patient’s dental occlusion to its preinjury state. Fractures without dislocation and no changes in the occlusion may be managed with maxillomandibular fixation alone, without the need for open surgery [14]. Maxillomandibular fixation can be achieved using several methods, including Erich arch bars, hybrid arch bars, intermaxillary fixation screws, and circum-mandibular wiring, as well as orthodontic brackets with hooks. However, most mandibular fractures require open reduction internal fixation (ORIF) for proper healing and restoration of the original maxillomandibular alignment. Various treatment approaches exist, depending on factors like the fracture site and the preferences of the surgeon. The patient’s age, comorbidities, dentition, fracture type, and the capabilities of the treatment center all contribute to the surgeon’s choice of the most suitable fixation and stabilization method [1,14,15]. ORIF can essentially be categorized into two types: load-bearing and load-sharing. Load-bearing osteosynthesis refers to a construct designed to bear 100% of the functional load generated by the mandible. This ensures that the bone at the fracture site bears none of the load, effectively protecting it from masticatory forces during healing. This is often achieved using locking reconstruction plates. Load-bearing fixation is typically used in cases of defect fractures, comminuted fractures, condylar fractures, and fractures in severely atrophic mandibles. In contrast, load-sharing osteosynthesis involves a fixation setup where the functional load is distributed between the titanium miniplate and the bone at the fracture site [14,15].

Despite advances in imaging diagnostics, there is still a small percentage of fractures that might be unrecognized either by surgeons or radiologists. The aim of this study was to present a series of cases involving undiagnosed concomitant secondary fractures in the mandibular body during preoperative diagnostics, as well as to describe the “air sign” as an indirect indicator of a mandibular body fracture.

## 2. Materials and Methods

From November 2022 to November 2023, a total of 75 patients underwent surgical treatment for mandibular fractures at the Department of Cranio-Maxillofacial Surgery, Jagiellonian University, Cracow. Among these cases, there were 64 (85.3%) isolated mandibular fractures and 11 (14.7%) panfacial fractures. In 3 (4%) cases, fractures of the body of the mandible were missed by the radiologists and surgeons before surgical treatment. Misdiagnosed fractures of the mandibular body were found in control CBCT after surgery in two cases and by clinical examination during surgery in one case. The failure to identify all fracture lines led us to conduct a thorough analysis of misdiagnosed cases. The retrospective analysis of preoperative diagnostic imaging (multi-planar reconstruction—MPR—of one CT and two CBCTs) revealed that no direct signs of fracture were visible. Only air accumulation in soft tissues near the misdiagnosed mandibular body fracture was observed, which was referred to as the “air sign”.

This study was approved by the institutional review board (No: 1072.6120.230.2021). General informed consent from the University Hospital in Cracow was obtained from all subjects involved in the study.

## 3. Results

### 3.1. Case 1

A 17-year-old female patient was admitted due to a fracture of the right angle of the mandible with an impacted wisdom tooth in the fracture line. The injury occurred as a result of a horse-riding accident. The patient lost consciousness without vomiting and alcohol consumption before the accident. The CT scans revealed minor bloody contusions adjacent to the right frontal lobe and a mandibular fracture of the right angle, with tooth 48 located along the fracture line (Figure 1). Intracranial injuries were qualified for conservative treatment after neurosurgical consultation.

Load-sharing ORIF was performed under general anesthesia, including extraction of tooth 48 from the fracture line. In the control CBCT scans, an additional fracture was found on the left side of the mandibular body (Figure 2). The patient did not agree to another surgery, and conservative treatment with maxillomandibular fixation (MMF) screws were applied with uneventful healing. The patient had the miniplate removed under local anesthesia three months after ORIF. Currently, the patient is scheduled for bimaxillary surgery due to hemifacial elongation. 

### 3.2. Case 2

A 19-year-old male patient was admitted due to a fracture of the right angle of the mandible. The injury occurred resulting from a fall (slipping while exiting a store). The injury did not lead to a loss of consciousness and the patient remembered the circumstances of the accident. The patient did not consume alcohol and denied experiencing either nausea or vomiting. The CBCT scanning before surgery revealed a mandibular fracture involving the right angle, with impacted tooth 48 located along the fracture line (Figure 3). 

Load-bearing ORIF was performed under general anesthesia, including extraction of tooth 48 from the fracture line. In the control CBCT scans, an additional fracture was found on the left side of the mandibular body (Figure 4). The second ORIF of the body of the mandible was performed under general anesthesia. The healing process was uneventful. 

### 3.3. Case 3

A 38-year-old female patient was admitted due to a fracture of the right condylar process of the mandible with dislocation. The injury occurred in the morning due to orthostatic hypotension while on holiday. The injury resulted in the loss of consciousness, although the patient remembered the circumstances of the accident. The patient did not consume alcohol before the accident. The CBCT before surgery revealed a dislocated fracture of the right condyle of the mandible (Figure 5). 

Load-bearing ORIF of the right mandibular condyle was performed with a 3D miniplate via an extraoral (transparotid) approach. During the surgery clinical examination revealed pathological movements of the body of the mandible. Intraoral revision of the mandibular shaft was performed, which revealed a symphysis fracture. Load-bearing ORIF of the symphysis fracture was performed at the same time (Figure 6). The healing process was uneventful.

A retrospective thorough analysis of the diagnostic imaging (multi-planar reconstruction—MPR—of one CT and two CBCTs) before surgery revealed only air collection, “air sign”, in the soft tissue near to the misdiagnosed fracture of the body of the mandible in each case (Figure 1a,b, Figure 3a,b and Figure 5a). On the other hand, 3D reconstruction of the CT/CBCT scans before surgery did not reveal a mandibular body fracture on the opposite site (Figure 1c and Figure 3c).

## 4. Discussion

The mandible is the only movable bone in the maxillofacial region and is exposed to external forces that can lead to fractures. Properly diagnosing a mandibular fracture and the subsequent treatment qualification are crucial for therapeutic success. There are several symptoms that may indicate a mandibular fracture. Starting with the clinical assessment provides information on patients susceptible to mandibular fractures and necessitating radiological imaging [16]. The findings from a physical examination play a crucial role in the diagnostic process of mandibular fractures. However, they frequently demonstrate limited sensitivity while tending to exhibit high specificity [16,17]. For instance, in a prospective study involving 119 patients, high specificity was observed for malocclusion (96%), facial asymmetry (96%), crepitus (96%), and sublingual hematoma (96%) [17]. The study by Rozema et al. also confirmed the high specificity of clinical examination in detecting mandibular fractures. In terms of extra-oral assessment, specificities were notable for swelling (88.6%), laceration (69.4%), mouth-opening restriction (90.2%), mandibular movement pain (87.6%), and auditory canal bleeding (96.1%). In intra-oral assessment, specificities included malocclusion (92.8%), tooth mobility or luxation (94.5%), tooth avulsion (99.1%), sublingual hematoma (93.4%), and gingival or mucosal laceration (89.5%). Palpation findings also exhibited high specificity, with the angular compression test showing pain (98.6%), the chin axial pressure test displaying pain (99.1%), palpable step-off presenting (99.1%), and inferior alveolar nerve paresthesia being observed (99.5%) [16].

In addition to physical examination and radiological assessments, the tongue blade test (TBT) serves as a sensitive screening tool for mandibular fractures. The TBT is a cost-effective tool widely available in emergency departments globally. Various studies have investigated the diagnostic accuracy of the TBT for mandibular fractures [18,19,20]. According to Alonso and Purcell, the TBT demonstrated a sensitivity of 97.5% and a specificity of 63.5% in detecting mandibular fractures [20]. The combination of clinical signs and the tongue blade test (TBT) has the potential to achieve accuracy comparable to radiological imaging. This combination might serve as a valuable clinical decision tool, helping to rule-out patients without mandibular fractures. The validation of such a clinical decision tool not only reduces unnecessary radiation exposure but also minimizes the inefficient use of time and resources in the emergency department [19].

Due to recent advancements in computed tomographic imaging technology, maxillofacial trauma surgeons now have access to new diagnostic tools. Since its introduction, computed tomographic imaging has significantly enhanced the radiographic visualization of the craniofacial skeleton when compared to plain film techniques [21]. However, in low-income and developing countries where equipment costs are a concern, conventional two-dimensional radiological diagnosis remains the predominant method for identifying mandibular fractures. Nonetheless, errors in interpreting radiographs could result in misdiagnoses and unfavorable consequences. Panoramic radiography, a two-dimensional imaging technique for mandibular fractures, is usually limited to isolated lesions. 

Many high-volume trauma centers prefer the use of two-dimensional computed tomographic scans due to factors such as accessibility, reasonable cost, the co-imaging capabilities of adjacent and remote traumatic body sites, and diagnostic familiarity. In some centers, preoperative diagnosis is also based on cone-beam computed tomography (CBCT) examination. Additionally, manipulations of two-dimensional computed tomographic scan data enable the generation of multiplanar reconstructions in sagittal and coronal views [22]. Nonetheless, CT/CBCT scans exhibit no overlap among distinct anatomical structures. Therefore, when dealing with multiple facial fractures or comminuted fractures, CT/CBCT scanning should be the preferred diagnostic tool over panoramic radiography for a more precise identification of the fracture lines [2]. CT/CBCT scans play a crucial role in identifying the location and direction of the fracture line, the degree and direction of displacement, and the degree of depression and rotation of bony fragments [23].

Recently, new tools for three-dimensional imaging, exemplified by software like OsiriX MD (Pixmeo SARL, Bernex, Switzerland) or RadiAnt Dicom Viewer (Medixant, Poznań, Poland), in conjunction with picture archiving and communication systems, have made three-dimensional computed tomographic imaging widely accessible to clinicians. These software applications simplify the process of generating three-dimensional computed tomographic models from a series of two-dimensional computed tomographic images on a personal workstation [21]. Among the reported drawbacks of 3D CT/CBCT images, there is insufficient resolution in specific craniofacial regions and a substantial presence of artifacts observed, especially in CBCT. This study, similarly to other studies, has suggested that depending solely on 3D CT may result in false-negative outcomes for non-displaced fractures [21,24,25].

CT/CBCT imaging is also effective in detecting air bubbles in the soft tissues of the head and neck, which can indirectly indicate fractures in multiple structures. Notably, the presence of an “air sign” in the orbital area typically signifies fractures of the orbital walls [26]. In cases of pneumocephalus, there is an implication of a skull base fracture [27]. Additionally, the detection of air near the styloid process of the temporal bone can be a marker of its fracture [28]. Additionally, the presence of air in the upper and middle facial region is indicative of fractures in the maxillary sinus walls, often seen in zygomatico-maxillary complex fractures or Le Fort fractures, as well as in fractures of the frontal sinus walls. The “air sign” described in this article represents an additional indirect indicator that may suggest a fracture of the mandibular body. The presence of air in soft tissues arises from the disruption in the continuity of the mucous membrane, gingiva, or dental pocket caused by stretching and crushing forces, leading to a fracture. In the area of injury and fracture, the deformation of the mandible generates negative pressure within the soft tissues, drawing air into the surrounding areas. It should be noted that air in the soft tissues around the mandible, visible in CT scans, can also originate from a facial skin wound (Figure 7). 

In such cases, the “air sign” may represent a false positive, thereby making a clinical examination essential for the accurate diagnosis of a potential mandibular body fracture. A false-positive “air sign” can also arise from the presence of air bubbles in the patient’s saliva, found in the vestibule of the oral cavity and the sublingual area, as well as from accompanying fractures such as a zygomatico-maxillary complex or Le Fort fractures with extensive subcutaneous emphysema. It is also important to emphasize that in old fractures of the body of the mandible, the “air sign” will not be visible on a CT/CBCT scan. Only imaging studies performed immediately after the injury will be able to detect the “air sign”, as in other cases, the air will have been reabsorbed into the soft tissues. However, given that over 50% of mandibular fractures involve more than one fracture line, in instances where the “air sign” is observed in the CT/CBCT scan around the mandibular body and when there are no skin wounds or other fractures involving the upper and middle face, the possibility of an additional fracture in the mandibular body should be considered. Therefore, after performing ORIF, it is essential to conduct a clinical evaluation during the surgical procedure for any pathological mobility of the mandibular body (teeth), which could indicate a previously undiagnosed accompanying mandibular body fracture. During ORIF, an accompanying undiagnosed mandibular body fracture may shift slightly and become visible during a follow-up radiological examination (Figure 2 and Figure 4). In the event that an additional fracture is detected after the ORIF procedure, it becomes necessary to perform another ORIF surgery, which prolongs the patient’s hospital stay, increases the cost of treatment, raises the risk of complications during the healing period, and heightens the risk associated with another general anesthesia. Additionally, an undiagnosed mandibular fracture before the ORIF procedure can undermine the competence of both the radiologist and the surgeon in the patient’s opinion and may lead to legal litigations.

Advancements in technology are reshaping numerous aspects of our society and industries, particularly within healthcare. Cutting-edge digital tools like computer-aided design/manufacturing, rapid prototyping, 3D bioprinting, augmented/virtual reality, artificial intelligence (AI) with deep learning models, and “omics” analysis are progressively finding applications across diverse medical fields, serving diagnostic and therapeutic functions [29,30]. Deep learning, a branch within machine learning, utilizes multi-layered artificial neural networks to comprehend data representations with various levels of abstraction [31]. These algorithms independently extract hierarchical features from intricate data, fine-tuning weighted parameters to enhance learning and reduce prediction errors. Within the spectrum of deep learning tools, the convolutional neural network (CNN) stands out as a widely embraced category of artificial neural networks proficient in tasks such as automated detection, segmentation, and classification of intricate patterns in both 2D and 3D images [29]. Radiomics, a developing field in quantitative imaging connected to machine learning, can quantify textural details from specific regions within digital diagnostic images. This process involves mathematically extracting signal intensity distributions and pixel/voxel interrelationships that escape human visual perception [32]. Overall, there exists a connection and mutual enhancement between AI and radiomics. Deep learning models utilize both panoramic radiography and CT scans to identify mandibular fractures [2,33,34,35,36,37]. According to the study by Wang et al., the CNN model achieved an accuracy ranging from 93.87% to 98.28% in detecting mandibular fractures across nine subregions [34]. In this study, the accuracy of the CNN model refers to mandibular fractures that were previously identified by surgeons, who themselves may also make diagnostic errors. Nevertheless, there is consistently a subset of patients in whom clinicians have failed to detect mandibular body fractures. This was evident in instances from the current study, where the accompanying fracture line remained unnoticed in the CT/CBCT scans. Mistakes in identifying fractures by humans frequently arise from missing a fracture due to the demanding schedules of radiologists/surgeons and fatigue, given that radiological evaluations are often carried out during night shifts. Furthermore, the experience of the individual interpreting the radiological study is crucial, and deep learning models aim to eliminate errors in radiological diagnosis arising from human imperfections.

The main limitation of the study is the small sample size. This was due to the fact that undetected fractures of the mandible body prompted the researchers to look for other indirect radiological signs that may indicate a fracture of the mandible body upon CT/CBCT examination. As mentioned earlier, the presence of an “air sign” is subject to a a high false-positive examination error. A prospective study is planned to evaluate the presence of the “air sign” in soft tissue and concomitant mandibular body fracture with an evaluation of sensitivity, specificity, and positive and negative predictive values. Only patients with a CT/CBCT scan within 3 days of injury will be eligible for the study. 

## 5. Conclusions

In summary, the “air sign” in a CT/CBCT scan may serve as an additional indirect indication of a fracture without dislocation in the mandibular body. Despite the fact that it carries a high risk of false-positive results, its presence should prompt the surgeon to conduct a more thorough clinical examination of the patient under general anesthesia after completing the ORIF procedure in order to rule out additional fractures. Future research will need to focus on examining the detection of mandibular fractures through a combination of panoramic radiography and CT/CBCT scans. Also, considering indirect signs such as the “air sign” will help deep learning and radiomics models to better diagnose mandibular fractures. 

## Figures and Tables

**Figure 1 diagnostics-14-00362-f001:**
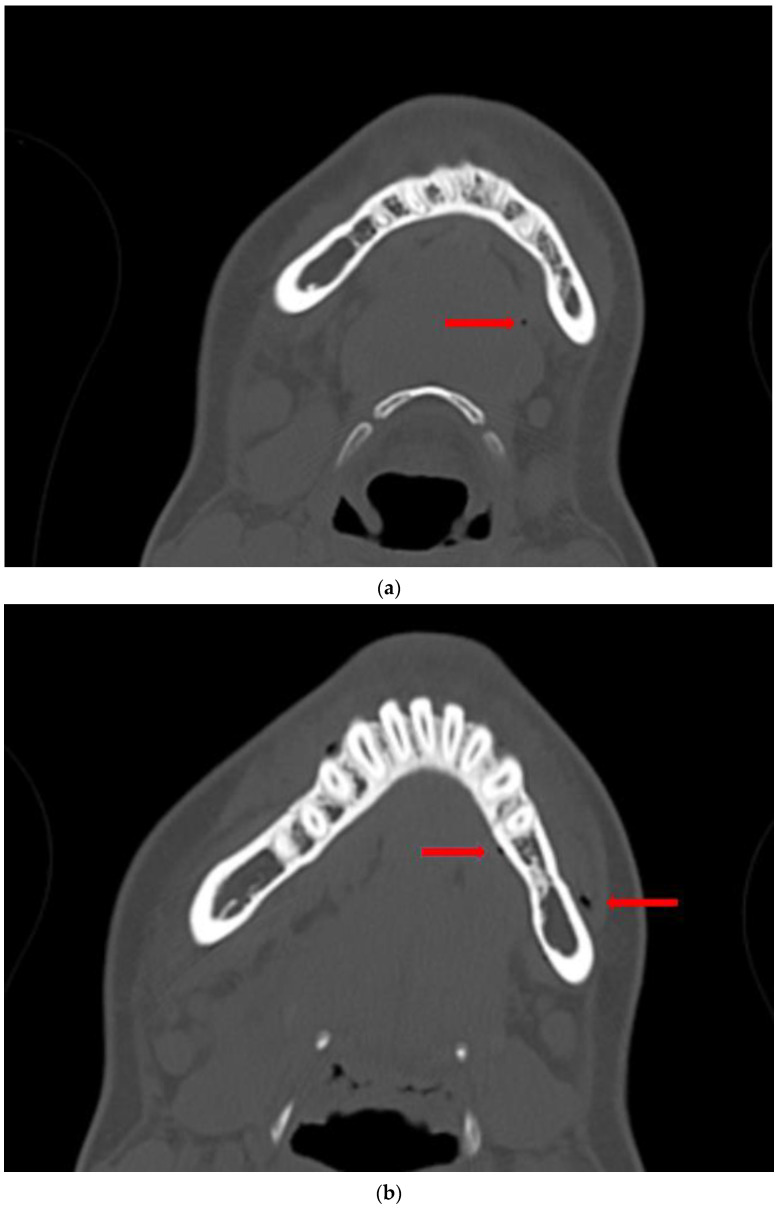
CT scans without contrast of the patient (Case 1) before surgery: axial view showing a collection of air in the soft tissues, identified as an “air sign” (red arrows), adjacent to the left side of the mandibular body (**a**,**b**); 3D reconstruction of the facial skeleton revealing a fracture at the right angle of the mandible, with no signs of fracture in the mandibular body on the left side (**c**).

**Figure 2 diagnostics-14-00362-f002:**
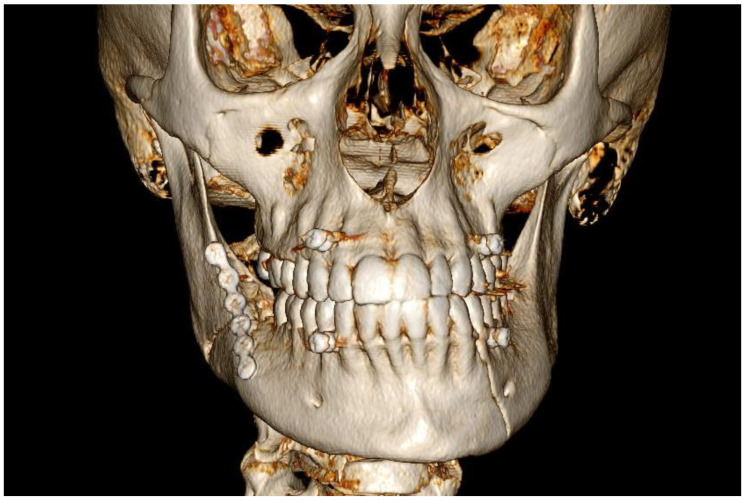
Postoperative control CBCT, with 3D reconstruction of the facial skeleton, following load-sharing ORIF of the right angle of the mandible revealed a fracture in the left mandibular body, which was not evident on the pre-ORIF 3D CT scan. (Case 1).

**Figure 3 diagnostics-14-00362-f003:**
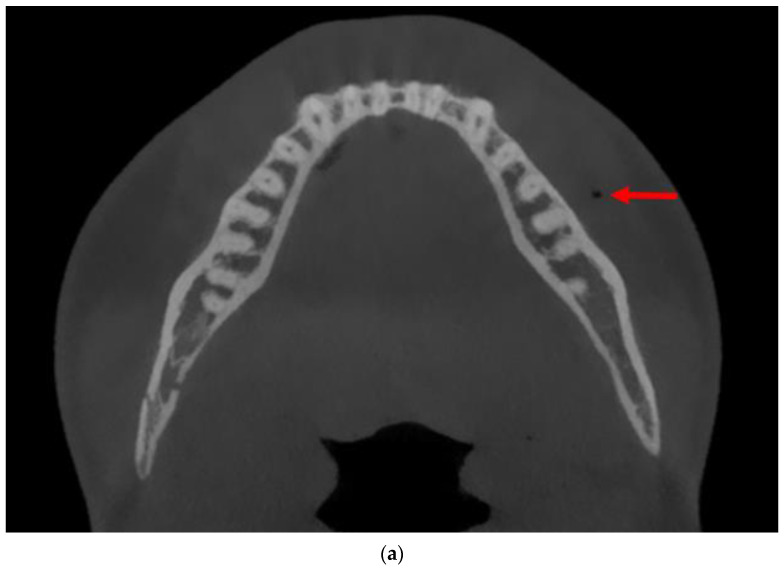
CBCT scans of the patient (Case 2) before surgery: Axial view showing a collection of air in the soft tissues, identified as an “air sign” (red arrows), adjacent to the left side of the mandibular body (**a**,**b**). However, “air sign” can also arise from the presence of air bubbles (especially close to the mandibular body) in the patient’s saliva, found in the vestibule of the oral cavity and the sublingual area (**b**); 3D reconstruction of the facial skeleton revealing fracture of the right angle of the mandible, with no signs of fracture in the mandibular body on the left side (**c**).

**Figure 4 diagnostics-14-00362-f004:**
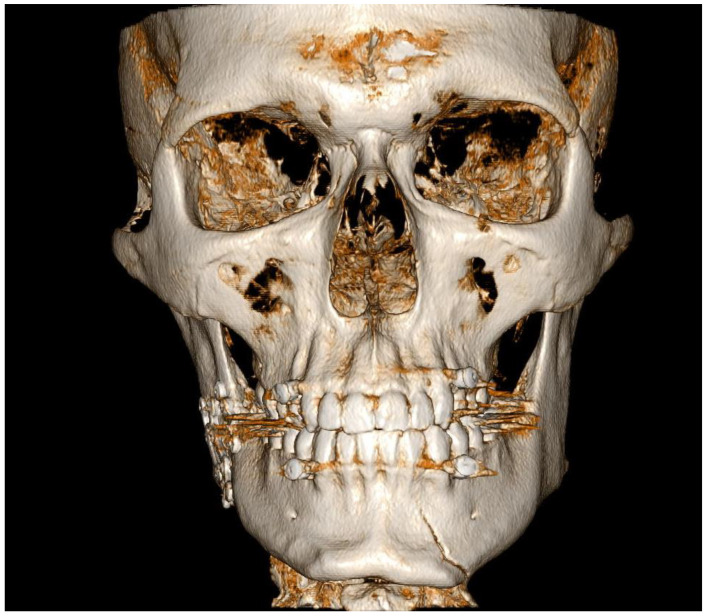
Postoperative control CBCT, with 3D reconstruction of the facial skeleton, following load-bearing ORIF of the right angle of the mandible revealed a fracture in the left mandibular body, which was not evident on the pre-ORIF 3D CBCT scan (Case 2).

**Figure 5 diagnostics-14-00362-f005:**
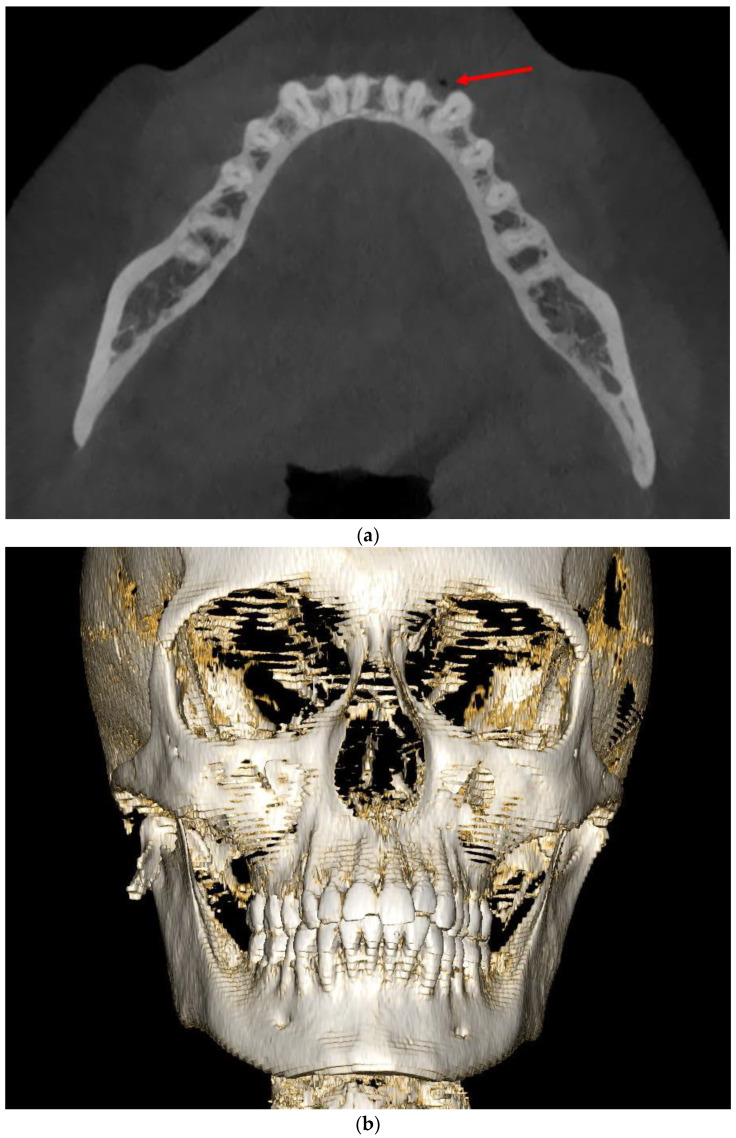
CBCT scans of the patient (Case 3) before surgery: axial view showing a collection of air in the soft tissues, identified as an “air sign” (red arrow), adjacent to the left side of the mandibular symphysis (**a**); 3D reconstruction of the facial skeleton revealing a fracture at the base of the mandibular condyle, without signs of fracture in the mandibular symphysis (**b**).

**Figure 6 diagnostics-14-00362-f006:**
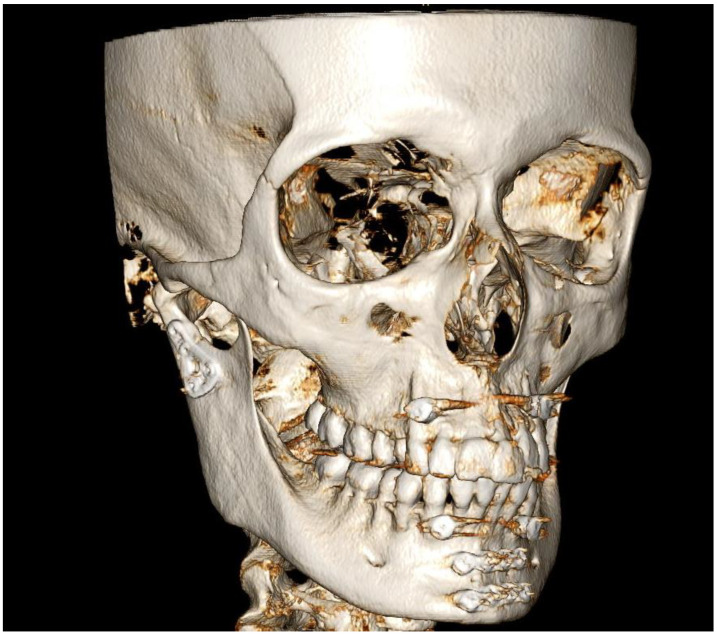
Postoperative CBCT with 3D reconstruction of the facial skeleton following load-bearing ORIF of the right mandibular condyle and symphysis fracture, which was revealed during clinical examination at the same time of surgery (Case 3).

**Figure 7 diagnostics-14-00362-f007:**
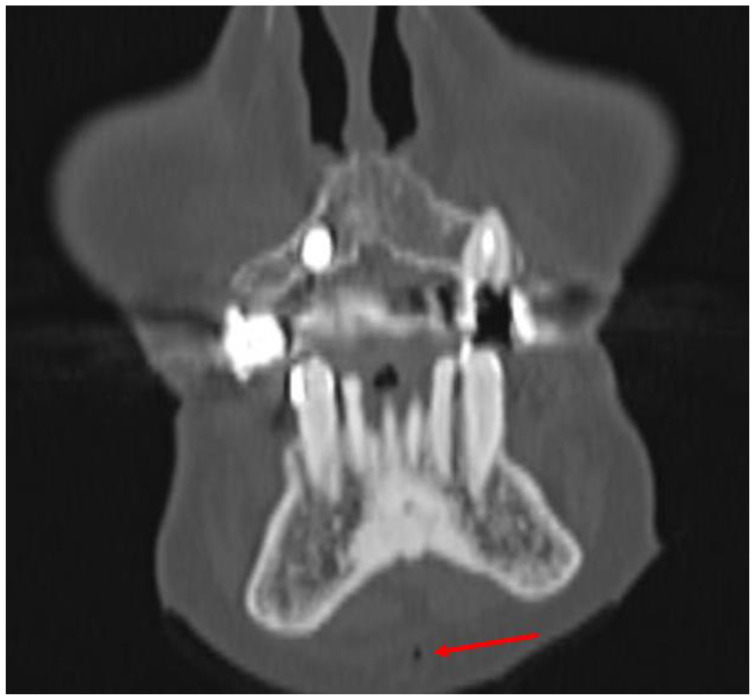
CBCT scans of the patient before surgery revealed a false-positive “air sign”: In the frontal view scan, a collection of air in the soft tissues (red arrow) was observed adjacent to the mandibular symphysis in a patient with a singular left-mandibular condyle fracture and a wound in the chin region. Clinical examination during surgery and control CBCT after load-bearing ORIF revealed no mandibular symphysis fracture.

## Data Availability

The data presented in this study are available on request from the corresponding author. Data is not publicly available due to privacy.

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
