# Peer review of "“Air Sign” in Misdiagnosed Mandibular Fractures Based on CT and CBCT Evaluation"

_diagnostics, 2024, doi:10.3390/diagnostics14040362_

Round 1

Reviewer 1 Report

Comments and Suggestions for Authors

This manuscript was a retrospective study to suggest that the "air sign" may be useful to avoid missing mandibular body fractures in CT/CBCT.

In 75 patients who underwent ORIF for mandibular fractures, a fracture was evident postoperatively in the mandibular body at another site in 4% of cases. The authors conclude that the presence of an "air sign" in the same region on preoperative CT/CBCT may be one finding to avoid missing fractures that were not evident preoperatively.

There were no reports that has ever suggested the usefulness of "air sign" for mandibular fractures, which is novel and very interesting. So, I think the conception of this study is excellent.

However, I have some concerns that should be addressed regarding the some issues including significant one.

Major:

1.     Page 2, Introduction, "air sign" needs to be mentioned. In particular, it is necessary to describe whether there are any reports of mandibular fractures.

2.     Page 3, Materials and Methods, the definition of "air sign" should be described.

3.     Page 3-, Results, the red arrows in figure are very unclear whether it is a resolution issue or not. It should be improved.

4.     Is the "air sign" on CT/CBCT (especially figure 1b) really an air caused by a fracture, i.e., can it be differentiated from an air in the area corresponding to the oral vestibule or floor of the mouth?

5.     Page 10, line 199-, Results, in all three cases, the postoperative 3D construction images are clear, but the preoperative 3D construction images are unclear. Although it is argued that the fracture was not obvious preoperatively, a fracture line can be seen on close examination. Could it be that the preoperative CT/CBCT slice width is just too large?

6.     Page 10, line 204-, Discussion, authors need to first state the summary of the results obtained in this study. In addition, the content is redundant and there is almost no description of how the results compare with the results of this study. I believe that it is unnecessary to include statements that do not supplement this study.

7.     Page 13, line 334-, although only 4% in this study, if there is an air sign, even if there is no obvious fracture line preoperatively, should we make an incision open the area where the air sign was present after ORIF to check for a fracture?

Minor:

1.     Page 1, abstract, (1)..., etc. are unnecessary.

2.     Page 1, line 31-39, Introduction, is the difference in gender relevant in this study?

3.     Page 2, line 88-95, Introduction, in this study, is it necessary to mention load-bearing or load-sharing?

4.     Page 3, line 105-106, Materials and Methods, "facture", shouldn't it be "fracture"?

5.     Page 4, line 134, Page 7, line 163, Results, the text says "load-sharing ORIF", but Page 5, Figure 2, Page 8, Figure 4 Figure legend says "load-bearing". Which is correct?

6.     Page 10, line 189, 192, Results, ORIF is performed for fractures in the condyle and symphysis, but it is described as load-bearing. In this case, isn't it load-sharing?

Comments on the Quality of English Language

In the Review.

Author Response

We thank the Reviewer for a very thorough evaluation of our manuscript. Excellent points have been raised that can improve our manuscript. We have tried to address each point raised with appropriate modifications, which can be viewed with the track changes throughout the manuscript. We are glad to address further issues if needed.

Please find attached our response to the comments made by the Reviewer.

Reviewer 2 Report

Comments and Suggestions for Authors

The missed diagnosis of mandibular fractures can pose considerable risks and harm to patients. Fortunately, recent advancements in imaging technology, coupled with the widespread adoption of CT scans, have led to a notable decrease in such missed diagnoses. Nevertheless, this remains a significant issue deserving of attention. In this manuscript, the authors present three cases of mandibular fractures that were initially missed and propose that the presence of an "air sign" could potentially aid in the accurate diagnosis of these fractures, thereby offering a valuable new insight to the medical community. However, there're still several issues which should be addressed.

Firstly, the limited sample size raises questions about the reliability and generalizability of the "air sign" as a diagnostic tool. Additional prospective studies are necessary to validate its effectiveness in a broader context. It would be beneficial for the authors to expand upon this aspect in their discussion section.

Secondly, the utility of the "air sign" appears to diminish in the case of old fractures. This important caveat should also be explicitly addressed in the discussion to avoid any potential misunderstandings.

Lastly, a significant portion of the manuscript's content appears to be tangential to its main focus. Specifically, the discussions on etiology and fracture treatment methods in the introduction section seem unrelated to the core theme of the manuscript. Similarly, the discussion section contains a substantial amount of extraneous material that detracts from the central argument. It is advisable for the authors to streamline their manuscript by deleting these non-essential parts and reducing the overall word count by at least 30% to enhance its coherence and readability.

In summary, while the manuscript sheds light on a potentially valuable diagnostic indicator for mandibular fractures, several issues need to be resolved to strengthen its clarity and focus. It is crucial for the authors to carefully consider these recommendations to optimize the impact and readability of their work.

Author Response

(The authors gave the same response as above.)

Round 2

Reviewer 1 Report

Comments and Suggestions for Authors

Thank you for your diligence and response to my questions.